# High Homocysteine Levels Are Associated with Cognitive Impairment in Patients Who Recovered from COVID-19 in the Long Term

**DOI:** 10.3390/jpm13030503

**Published:** 2023-03-10

**Authors:** Pinar Oner, Seda Yilmaz, Serpil Doğan

**Affiliations:** 1Department of Microbiology, Elazig Fethi Sekin City Hospital, Elazig 23100, Turkey; 2Department of Psychiatry, Elazig Fethi Sekin City Hospital, Elazig 23100, Turkey; 3Department of Neurology, Elazig Fethi Sekin City Hospital, Elazig 23100, Turkey

**Keywords:** COVID-19, homocysteine, cognition, depression, long term

## Abstract

In this study, we measured the levels of depression and cognition in people recovering from COVID-19. Moreover, we aimed to investigate the relationship between depression and cognition levels by measuring homocysteine concentrations. It included 62 people recovering from COVID-19 (at least 3 months after positive RT-PCR) and 64 people without COVID-19 (control group). At first, the homocysteine levels of participants were measured. Beck Depression Inventory (BDI) and Montreal Cognitive Assessment (MoCA) were performed to collect data. Homocysteine levels of the group recovering from COVID-19 (x^−^ = 19.065 µmol/L) were higher than the control group (x^−^ = 11.313 µmol/L). There was no significant difference between the groups for BDI scores. The MoCA scores of the group recovering from COVID-19 (x^−^ = 20.774) were lower than the control group (x^−^ = 24.297). There was a negative high (r = –0.705, *p* < 0.001) correlation between homocysteine levels and MoCA scores. Linear regression analysis is shown to be significant, and the MoCA explanatory value of the variables in the model is 58.6% (*p* < 0.0001). A 1 µmol/L observed increase in homocysteine level constituted a risk for a 0.765-point decrease in MOCA scores. In patients recovering from COVID-19, early interventions to high homocysteine levels may prevent cognitive impairments that may persist in the long term.

## 1. Introduction

The COVID-19 pandemic is one of the biggest health problems in recent history. This pandemic, which first appeared in China, quickly affected the world, and millions of cases and deaths were reported [1]. The clinical picture can be seen with many symptoms such as fever, sore throat, weakness, muscle aches, loss of smell, and cough in patients [2]. Many different clinical manifestations have been discovered so far, but we know very little about the long-term effects of this disease. The long-term effects of COVID-19, called ‘Long COVID’, have attracted attention recently [3]. In a recent longitudinal study, some morphological changes were observed in the brains of those with COVID-19 [4]. This result is exciting to understand the psychiatric and neurological consequences that may occur in patients with this infection and to plan new studies.

COVID-19 infection can affect the central nervous system in different ways. The developing immune response to SARS CoV-2, which settles in the respiratory system, may lead to an increase in cytokines, chemokines, and immune cells that increase neuroinflammation in the brain. Even if SARS-CoV-2 is rare, it can reach the nervous system directly. SARS-CoV-2 may generate an autoimmune response against the nervous system. Activation of latent herpesviruses such as Epstein–Barr virus during COVID-19 infection can trigger neuronal damage. SARS-CoV-2 can disrupt blood flow in nerve cells by triggering the formation of cerebrovascular and thrombotic diseases. This may also disrupt the blood–brain barrier and increase the severity of neuroinflammation and ischemia. In addition, pulmonary and multi-organ dysfunction disorders caused by severe COVID-19 can lead to conditions that can negatively affect neural cells by leading to nerve cells hypoxemia, hypotension, and metabolic disorders [5].

Homocysteine is not found in the diet; it is an amino acid produced from methionine. Methionine and homocysteine are both precursors of each other. The ubiquitous methionine cycle is an essential part of body metabolism [6]. Homocysteine can be involved in two separate metabolic pathways: transsulfuration and methylation pathway. A high level of homocysteine in the blood is called hyperhomocysteinemia, and this condition has been associated with some diseases [7]. Hyperhomocysteinemia is associated with stroke, heart attack, and cardiovascular disease. Hyperhomocysteinemia causes endothelial damage in vessels and causes deterioration of existing hemostasis. It also contributes to the development of inflammation [8]. Other proinflammatory factors, such as homocysteine and interleukin-6 (IL-6), C-reactive protein (CRP), and alpha-1-antimotrypsin, have been associated with neuroinflammation and cognitive decline. Moreover, homocysteine has been reported as an independent factor in the impairment of information processing, general cognitive function, and fluent intelligence. The combination of high homocysteine and increased inflammation can be used as an indicator of cognitive impairment [9]. There is increasing evidence of an association between high homocysteine levels and depression [10]. Hyperhomocysteinemia is closely associated with neurodegenerative diseases as well as poor cognitive performance [11].

It has been observed that there is an increase in psychiatric diseases such as depression in COVID-19 patients [12]. COVID-19 infection can cause cognitive decline [4]. Additionally, among COVID-19 patients, those with worse outcomes have been found to have higher homocysteine levels [7]. Studies on homocysteine in COVID-19 have focused on the cardiovascular system. This study, however, includes a neuropsychiatric approach. In addition, it is new research for ‘Long COVID’. In this study, we measured the levels of depression and cognition in people who had previously recovered from COVID-19. Furthermore, we aimed to investigate the relationship between depression and cognition levels and homocysteine by measuring homocysteine concentrations. 

## 2. Materials and Methods

### 2.1. Inclusion and Exclusion Criteria

One hundred twenty-six people who were admitted to the outpatient neurology clinics of Elazig Fethi Sekin City Hospital were included in the study. Previous data of 62 people in our hospital’s microbiology laboratory had at least 1 positive reverse transcription polymerase chain reaction (RT-PCR) test for severe acute respiratory syndrome coronavirus 2 (SARS-CoV-2). Previous data of 64 people in the control group had no history of COVID-19 and no positive RT-PCR test for SARS-CoV-2. For the recovered COVID-19 group, only those who had recovered from COVID-19 infection once were included in the study; those who had recovered from repeated COVID-19 infection were not included in the study. Homocysteine levels of all participants were measured at the time of admission. Although there are different definitions for ‘Long COVID’ or ‘post-COVID-19 syndrome’, the most common definition deals with showing symptoms lasting more than 3 months from the first symptom onset [3]. In this study, only those patients who had received a positive RT-PCR test for COVID-19 at least 3 months earlier were included in the group of those who had recovered from COVID-19. For both groups, those between the ages of 18 and 65 were included in the study. Since some psychiatric and cognitive disorders may occur in the patients in the intensive care unit in the long term [13], those who had been hospitalized in the intensive care unit and who had had severe COVID-19 infection were excluded from the study. Since homocysteine levels may be high in obese patients [14], only those patients with BMI < 30 were included in the study. Again, exclusion criteria for both groups were a history of cardiovascular system disease or thromboembolism, a history of chronic disease, mental retardation, a neurological disorder that may cause cognitive impairment, a history of psychiatric disease, alcohol-substance use, smoking, folic acid use, and continuous drug use. At the same time, the Beck Depression Inventory (BDI) and Montreal Cognitive Assessment (MoCA) results, which were evaluated at the time of the patient’s admission, were included in the study. All participants were provided with detailed information about the study, and then a written consent form was obtained. The principles of the Declaration of Helsinki were followed throughout the entire study. Our study was approved by the Elazig Firat University Clinical Research Ethics Committee (No: 2022/02-31).

### 2.2. Hematological Analysis

#### Homocysteine

Blood samples were taken from the antecubital vein (6 mL). Samples were collected and analyzed in vacuum tubes, including 15% K3 ethylene diamine tetraacetic acid-anticoagulation tubes (Sarstedt, Essen, Belgium). The levels of homocysteine were tested using the spectrophotometric method on an Abbott Architect c8000 (Abbott, Abbott Park, IL, USA) Chemistry System.

### 2.3. Data Collection Tools

#### 2.3.1. Beck Depression Inventory (BDI)

Beck et al. [15] developed this scale. Hisli [16] reported the Turkish validity and reliability study of the scale. It consists of 21 items in total. The participant is asked to give a score between 0 and 3 for each item. The total score obtained from all items reveals the severity of depression, and an increase in this score indicates an increase in the severity of depression.

#### 2.3.2. Montreal Cognitive Assessment (MoCA)

This is a scale used to assess mild cognitive impairment. Cognitive functions such as visuospatial and executive functions, naming, memory, attention, language, abstraction, and orientation are evaluated. A maximum of 30 points can be obtained from this scale. A score of less than 21 indicates cognitive impairment, while higher scores indicate better cognition. Nasreddine et al. [17] developed this scale, and Selekler et al. [18] performed Turkish validity and reliability study. 

### 2.4. Statistical Analysis

The data obtained in the research were analyzed using the SPSS 22.0 (Statistical Package for Social Sciences) for Windows program. Kurtosis and skewness values were examined to determine whether the research variables showed a normal distribution, and it was observed that they showed a normal distribution. Parametric methods were used in the analysis of the data. The difference of continuous variables according to the groups was analyzed with the t-test. Differences between the ratios of categorical variables in independent groups were analyzed with Chi-square and Fisher’s exact tests. Pearson’s correlation analysis was performed between the continuous variables of the study. Linear regression analysis was performed to determine the predictive parameters of MoCA scores. Receiver-operating characteristic (ROC) analysis was used to estimate the optimal cut-off values of homocysteine in patients who had recovered from COVID-19. *p* < 0.05 values were considered to indicate statistical significance.

## 3. Results

There was no significant gender difference between the groups (χ^2^ = 0.031; *p* = 0.501). It is seen that there are 31 (50.0%) men and 31 (50.0%) women in the group who had recovered from COVID-19, and 33 (51.6%) men and 31 (48.4%) women in the control group. There was no significant difference in terms of age between the group who had recovered from COVID-19 (39.742 ± 9.426) and the control group (40.516 ± 9.188). There was no significant difference in terms of the years of formal education between the group who had recovered from COVID-19 (7.048 ± 5.787) and the control group (6.063 ± 5.345). The mean time that passed after those who had recovered from COVID-19 received their positive RT-PCR test results was found to be 10.435 ± 5.180 (Table 1). Homocysteine levels differed significantly between the groups (t_(124)_ = 10.306; *p* < 0.001). Homocysteine levels (x^−^ = 19.065 µmol/L) of the group who had recovered from COVID-19 were found to be higher than the homocysteine levels (x^−^ = 11.313 µmol/L) of the control group (Table 1) (Figure 1). There was no significant difference between the BDI scores of the group who had recovered from COVID-19 (x^−^ = 4.274) and the BDI scores (x^−^ = 3.515) of the control group (Table 1). MoCA scores differed significantly between the groups (t_(124)_ = −5.137; *p* < 0.001). The MoCA scores of the group who had recovered from COVID-19 (x^−^ = 20.774) were lower than the MoCA scores (x^−^ = 24.297) of the control group (Table 1) (Figure 1). Comparison of homocysteine, MoCA, BDI levels between the two groups is shown in Appendix A.

The analyses of the correlation coefficients indicated there was a high negative correlation with a value of between the homocysteine levels and the MoCA scores in patients who recovered from COVID-19 (r = −0.705, *p* < 0.001) (Table 2) (Figure 2). There was a positive correlation between the MoCA scores and the age of the participants in patients who recovered from COVID-19 (r = 0.252, *p* = 0.048). There was a positive correlation between the MoCA scores and the years of formal education in patients who recovered from COVID-19 (r = 0.253, *p* = 0.048). Correlation relationships between other variables were not statistically significant (Table 2). 

There was a positive correlation between the MoCA scores and the years of formal education in the control group (r = 0.303, *p* = 0.015). There was a negative correlation between the homocysteine levels and the MoCA scores in the control group (r = –0.260, *p* = 0.038) (Table 3).

Linear regression analysis was performed on the patients who recovered from COVID-19. It is seen that this model is significant (F:15.836; df:5; *p* < 0.0001), and the MoCA explanatory value of the variables in the model is 58.6% (R^2^ = 0.586; *p* < 0.0001). It was observed that a 1 µmol/L increase in homocysteine level constituted a risk for a 0.765-point decrease in MOCA scores (Table 4). 

Linear regression analysis was performed in the control group. It is seen that this model is significant (F:16.527; df:4; *p* < 0.001), and the MoCA explanatory value of the variables in the model is 35.3% (R^2^ = 0.353; *p* < 0.001). It was observed that a 1 µmol/L increase in homocysteine level constituted a risk for a 0.594-point decrease in MOCA scores (Table 5). 

Linear regression analysis was performed in all participants. It is seen that this model is significant (F:28.021; df:4; *p* < 0.0001), and the MoCA explanatory value of the variables in the model is 48.1% (R^2^ = 0.481; *p* < 0.0001). It was observed that a 1 µmol/L increase in homocysteine level constituted a risk for a 0.693-point decrease in MOCA scores (Table 6). 

The ROC analysis demonstrated that homocysteine > 14 had 83.87% sensitivity and 82.81% specificity for predicting recovery from COVID-19 (AUC: 0.906, % 95 CI:0.841, 0.951; *p* < 0.0001; cut-off > 14) (Figure 3).

## 4. Discussions

In line with the results of our study, we found that homocysteine levels were higher in those who had recovered from COVID-19 than the control group. There was no significant difference in BDI scores between the groups, but those who had recovered from COVID-19 had lower MoCA scores. We found a highly negative correlation between homocysteine levels and MoCA scores in those who had recovered from COVID-19. In the regression analysis, a 1 µmol/L increase in homocysteine level constituted a risk for a 0.652-point decrease in MOCA scores.

In recent studies, it has been observed that the level of homocysteine is high in COVID-19 patients. In a retrospective study, homocysteine levels were found to be significantly higher in COVID-19 patients who did not survive [19]. Homocysteine has been proposed as a potential biomarker for cardiovascular risk in COVID-19 patients [20]. As it is known, there is a positive relationship between thromboembolism and homocysteine levels [21]. In our study, homocysteine levels were found to be significantly higher in patients who had recovered from COVID-19. For its viral RNA, SARS-CoV-2 transfers a methyl group from the host cell’s S-adenosylmethionine (SAM) to S-adenosylhomocysteine (SAH) and produces homocysteine through a series of metabolic processes [19]. This mechanism of SARS-CoV-2 that causes homocysteine production may be the reason for the high homocysteine levels that we found in our study in patients who had recovered from COVID-19. In addition, it is known that disruption of enzymes involved in the metabolism of B vitamins increases homocysteine concentration [22]. The reason for the high homocysteine concentration we found in our study may be due to a direct viral effect or a secondary mechanism of action causing these enzymes to deteriorate in those who had recovered from COVID-19. In addition, in our study, there was no relationship between the time that passed after being tested positive for COVID-19 and homocysteine levels. In other words, homocysteine levels remained high in those who had recovered from COVID-19 a long time ago. High homocysteine levels are associated with the MTHFR mutation. The most common single nucleotide polymorphism for MTHFR is the MTHFR C677T polymorphism. This is a common genetic cause of hyperhomocysteinemia [21]. Together with the more severe course of COVID-19 disease, MTHFR C677T polymorphism and hyperhomocysteinemia were thought to be associated [23,24]. MTHFR C677T polymorphism may be the reason why homocysteine levels of patients who had recovered from COVID-19 remained high despite the time that passed in our study.

Depressive symptoms have been reported in COVID-19 patients [10]. In a study, it was observed that patients hospitalized due to COVID-19 had increased depression rates in the next one-month follow-up [25]. A review of the post-COVID-19 syndrome concluded that there might be clinically significant depressive symptoms, but this does not mean that people with post-COVID-19 syndrome will have a higher depression rate than the general population [26]. In our study, there was no significant difference between the depression levels of patients who had recovered from COVID-19 and the control group. This result may be related to the time that passed between the active infection period of those who had recovered from COVID-19 in the sample of our study and the time when BDI was applied, which made us think that COVID-19 may not cause depression in the long term.

Cognitive impairment can be described as a common symptom of COVID-19 infection. Brain fog with cognitive decline has been found to be significantly associated with ‘Long COVID’ [27]. Many factors that have not yet been clarified have been proposed as the cause of cognitive decline in those who have recovered from COVID-19. These factors include hypoxia, respiratory failure, drugs used, a direct viral infection of the central nervous system, inflammation, endothelial damage, and cerebrovascular events [28]. In a previous study, cognitive impairments were seen in cases of those aged 64 years on average recovering from severe COVID-19 [29]. In another study, cognitive impairments were observed in cognitive function assessments of patients hospitalized for COVID-19 infection before discharge [30]. In our study, which included patients with an average age of 39 years and who had recovered from COVID-19 infection long before those patients in the previous studies, the MoCA scores of those who had recovered from COVID-19 were considerably lower than the control group. This result supported the negative effect of COVID-19 on cognitive functions. Those who were studied longer and those who were younger had higher MoCA scores. This is also an expected result because better cognitive performance is observed in those with younger ages and higher education levels [31]. In addition, MoCA scores were not associated with the time that passed after being tested positive for COVID-19. Those who recovered from COVID-19 longer ago also continued to have low MoCA scores. This result showed that cognitive impairment might continue even after a long period since the onset of the COVID-19 infection.

Hyperhomocysteinemia is a risk factor for neurodegenerative diseases. Many studies show that homocysteine plays a role in cognitive impairment, memory decline, and brain damage [32]. In our study, in which we think that prolonged cognitive impairment may be related to homocysteine, there was a highly negative correlation between homocysteine levels and MoCA scores in patients who had recovered from COVID-19. Those with higher homocysteine levels had lower MoCA scores, meaning they had more cognitive impairment. In addition, in the regression analysis, we found that the increase in homocysteine level poses a risk for the decrease in MOCA scores. There are some possible reasons for these results. Homocysteine is neurotoxic and may compromise the integrity of the blood–brain barrier [33]. Apart from that, homocysteine initiates a proinflammatory process and causes neurological dysfunction through oxidative stress. Oxidative stress caused by homocysteine can be caused by an increase in reactive oxygen species, inactivation of the nitric oxide synthase pathway, and lipid peroxidation, which forms in the brain by blocking the NMDA receptor [32]. Hyperhomocysteinemia is associated with thromboembolism and vascular damage [21]. Therefore, homocysteine may cause cognitive impairment by causing cerebrovascular events. Based on the results of our study, homocysteine, which was previously seen as a risk factor for neurodegenerative diseases and cognitive impairment, can be considered as one of the causes of cognitive impairment in the long term in patients who recovered from COVID-19.

Treatment of hyperhomocysteinemia varies according to the underlying cause. Despite different underlying causes, treatments with vitamin supplements containing B6, B12, and folic acid can be effective in reducing homocysteine levels [22]. Although the cause of high homocysteine levels in patients with COVID-19 is not fully known, it may be beneficial to use B vitamins in the treatment to prevent cognitive functions by reducing homocysteine levels in these patients in the long term.

There were some limitations in our study. The number of participants in the study was small. In order to generalize our results to society, new studies with a larger number of participants should be carried out. There were no detailed data in our study on the severity of infection of those who had recovered from COVID-19. New studies can be planned in which patients who have experienced mild, moderate, and severe COVID-19 infection are examined separately. In addition, while cognitive impairments are measured, impairments in different cognitive areas can be examined in more detail in new studies. It is not known which of the factors that may cause high homocysteine concentration, such as direct viral effect, gene mutation, and impaired vitamin metabolism, have an effect on patients with COVID-19. We recommend that further studies be conducted to elucidate the underlying causes. Thus, in patients with COVID-19, early interventions to high homocysteine levels will prevent cognitive disorders that may persist in the long term. The MoCA scale has some limitations. In addition to showing the significant and moderate correlations of MoCA subtest scores with the cognitive areas that are intended to be evaluated, the accuracy rate is poor in predicting cognitive impairment in their specific areas. Unlike neuropsychological test scores, the fact that MoCA subtest scores are not adjusted for age or education level may cause poor validity estimates. Though the MoCA subtest scores show cognitive impairment with a slightly above-average accuracy, they remain below the expected level for diagnostic screening in terms of health care. Physicians who want to minimize false positives or negatives may prefer to interpret the performance of a particular screening test based on its high specificity or high sensitivity in a cognitive area. However, in this case, it will not guarantee an accurate identification of cognitively impaired individuals and will not completely reflect the general cognitive ability model. Therefore, it is recommended to dispatch this information to a neuropsychological evaluation in cases where it is necessary for diagnostic purposes [34]. Many modern methods are available for detecting and diagnosing diseases. Combining the homocysteine level we have detected in the blood with modern methods in new studies may be an important scientific development [35,36,37,38].

## 5. Conclusions

In conclusion, homocysteine levels were higher and MoCA scores were lower in our study for those who had recovered from COVID-19. However, there was no significant difference in terms of BDI. As a very important finding, there was a high negative correlation between homocysteine and MoCA scores, and an increase in homocysteine level poses a risk for a decrease in MOCA scores. In our study, homocysteine levels remained high, and cognitive impairment continued despite the time that passed after COVID-19 positivity. We think that our study provides important data for ‘Long COVID’. Our findings suggest that early intervention is necessary for high homocysteine levels and cognitive impairment, which may persist for a longer period and tend to be chronic. Cognitive impairment that may occur in the long term in patients who have recovered from COVID-19 should be evaluated and homocysteine levels that may cause cognitive impairment should be measured. Until a new and more effective treatment is found, we believe that it will be beneficial to support cognitive functions by lowering homocysteine levels with vitamin B supplementation.

## Figures and Tables

**Figure 1 jpm-13-00503-f001:**
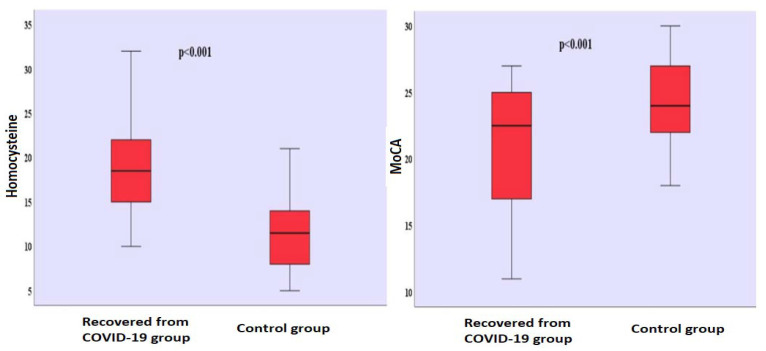
Comparison of homocysteine levels and MoCA scores between groups. MoCA: Montreal Cognitive Assessment.

**Figure 2 jpm-13-00503-f002:**
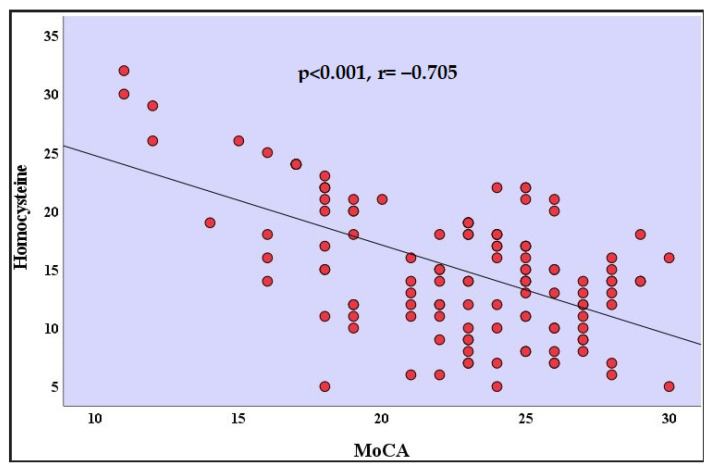
Correlation analysis between homocysteine and MoCA in the recovered from COVID-19 group. MoCA: Montreal Cognitive Assessment.

**Figure 3 jpm-13-00503-f003:**
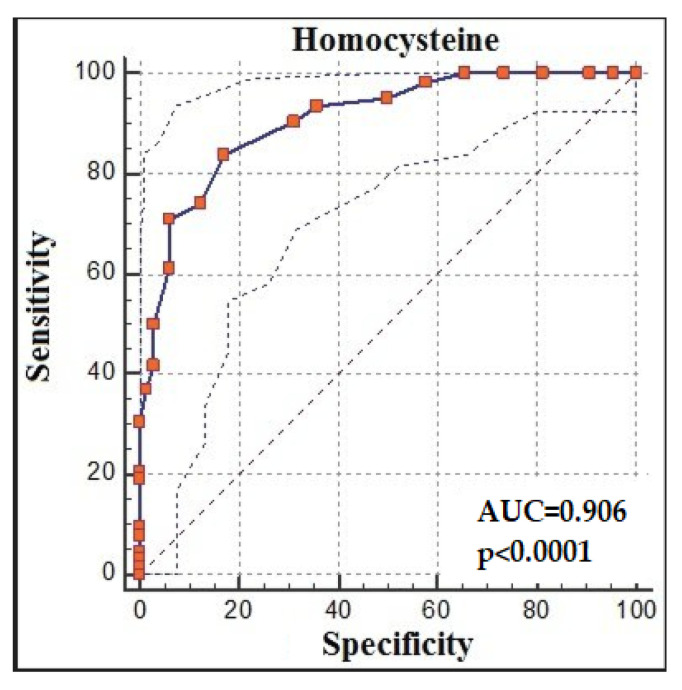
Homocysteine receiver-operating characteristic (ROC) analysis between groups.

**Table 1 jpm-13-00503-t001:** Sociodemographic and clinical data.

		Recovered from COVID-19 Group		Control Group		Total		*p*
		n	(%)	n	(%)	n	(%)	
Gender	Male	31	50.0%	33	51.6%	64	50.8%	χ^2^ = 0.031 *p* = 0.501
	Female	31	50.0%	31	48.4%	62	49.2%	
Socio- Economic Status	Low	34	55.8%	37	57.8	71	56.3%	χ^2^ = 1.176 *p* = 0.249
	Middle	17	27.4%	15	23.4	32	25.4%	
	High	11	16.8%	12	18.8	23	18.3%	
		Mean ± SD		Mean ± SD		t		*p*
Age		39.74 ± 9.42		40.51 ± 9.18		−0.46		0.642
Years of formal education		7.04 ± 5.78		6.06 ± 5.34		0.99		0.322
Homocysteine (µmol/L)		19.06 ± 4.68		11.31 ± 3.72		10.30		<0.001
BDI		4.27 ± 2.73		3.51 ± 2.61		1.59		0.115
MoCA		20.77 ± 4.47		24.29 ± 3.13		−5.13		<0.001
Vitamin B (ng/L)		184 ± 7.25		201 ± 10,43		−0.58		0.593
Folic acid (µg/L)		5.47 ± 0.71		6.19 ± 0.86		−1.21		0.275
The time that passed after positive RT-PCR test (Months)		10.43 ± 5.18		-		-		-

Chi-square analysis; independent groups t-test. BDI: Beck Depression Inventory; MoCA: Montreal Cognitive Assessment.

**Table 2 jpm-13-00503-t002:** Correlation analysis for homocysteine and MoCA in the recovered from COVID-19 group.

	Homocysteine (µmol/L)		MoCA	
	R	*p*	r	*p*
Age	0.247	0.053	0.252	0.048 *
Years of formal education	-	-	0.253	0.048 *
The time that passed after positive RT-PCR test (Months)	0.020	0.879	−0.071	0.581
MoCA	−0.705 **	<0.001	-	-

* *p* < 0.05; ** *p* < 0.001; Pearson’s correlation analysis. MoCA: Montreal Cognitive Assessment; RT-PCR: reverse transcription polymerase chain reaction.

**Table 3 jpm-13-00503-t003:** Correlation analysis for homocysteine and MoCA in the control group.

	Homocysteine (µmol/L)		MoCA	
	r	*p*	r	*p*
Age	−0.040	0.753	−0.077	0.547
Years of formal education	-	-	0.303 *	0.015
MoCA	−0.260 *	0.038	-	-

* *p* < 0.05; Pearson’s correlation analysis. MoCA: Montreal Cognitive Assessment.

**Table 4 jpm-13-00503-t004:** Linear regression analysis for MoCA in the recovered from COVID-19 group.

	Unstandardized Coefficients		Standardized Coefficients	t	*p*
	B	Std. Error	β		
(Constant)	34.388	2.399		14.336	<0.0001
Age	−0.054	0.043	−0.114	−1.265	0.211
Gender	−0.098	0.828	−0.011	−0.118	0.906
Years of formal education	0.218	0.070	0.282	3.134	0.003
Homocysteine (µmol/L)	−0.650	0.085	−0.681	−7.607	<0.0001
The time that passed after positive RT-PCR test (Months)	−0.043	0.078	−0.050	−0.559	0.578

Linear regression analysis, model *p* < 0.0001; R^2^ = 0.586. Dependent variable: MoCA. RT-PCR: reverse transcription polymerase chain reaction.

**Table 5 jpm-13-00503-t005:** Linear regression analysis for MoCA in the control group.

	Unstandardized Coefficients		Standardized Coefficients	t	*p*
	B	Std. Error	β		
(Constant)	30.216	1.897		15.926	<0.001
Age	−0.057	0.033	−0.125	−1.706	0.091
Gender	0.220	0.618	0.026	0.355	0.723
Years of formal education	0.107	0.056	0.141	1.912	0.058
Homocysteine (µmol/L)	−0.423	0.054	−0.574	−7.806	<0.001

Linear regression analysis, model *p*<0.001; R^2^ = 0.353. Dependent variable: MoCA.

**Table 6 jpm-13-00503-t006:** Linear regression analysis for MoCA in all participants.

	Unstandardized Coefficients		Standardized Coefficients	t	*p*
	B	Std. Error	β		
(Constant)	29.648	1.718		17.255	<0.0001
Age	−0.040	0.030	−0.088	−1.335	0.184
Gender	0.416	0.556	0.050	0.749	0.455
Years of formal education	0.175	0.050	0.231	3.497	0.001
Homocysteine (µmol/L)	−0.479	0.048	−0.650	−9.893	<0.0001

Linear regression analysis, model *p* < 0.0001; R^2^ = 0.481. Dependent variable: MoCA.

## Data Availability

The data presented in this study are available upon request from the corresponding author.

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
