# Peer review of "High Homocysteine Levels Are Associated with Cognitive Impairment in Patients Who Recovered from COVID-19 in the Long Term"

_jpm, 2023, doi:10.3390/jpm13030503_

Round 1

Reviewer 1 Report

In this study, the authors measured the levels of depression and cognition in people recovering from COVID-19. They aimed to investigate the relationship between depression and cognition levels by measuring homocysteine concentration. They found that homocysteine levels were higher and MoCA scores were lower in those who had recovered from COVID-19. Moreover, there was a high level of negative correlation between homocysteine and MoCA scores, and an increase in homocysteine level poses a risk for a decrease in MOCA scores. Overall, the findings are interesting but the authors need to improve their study specifically with their methods. Below are some specific suggestions for strengthening the manuscript.

1.     In this study, the sample size is small and may lead to imprecise results. If possible, more patients should be included in the analysis.

2.     It is known that disruption of enzymes involved in the metabolism of B vitamins increases homocysteine concentration. Detailed information on the history of taking folic acid and the B vitamins levels in the blood needs to be provided in those who had recovered from COVID-19 and the control group, especially the history of taking folic acid.

3.     Even though there was no statistically significant difference between the two groups in terms of age, gender, and education duration, I strongly suggest that the authors should take age, gender, education duration, and the time that passed after a positive RT-PCR test as covariates to regress out in evaluating the difference of homocysteine levels between the groups.

4.     Other demographic information in those who had recovered from COVID-19 and the control group, such as ethnicity, smoking history, alcohol consumption history, and socioeconomic status.

5.     Pearson's correlation analysis and linear regression analysis should also be performed in the control group. And I'm curious whether the results of the linear model regression for MoCA scores and homocysteine level would still be significant if age, gender, education duration, and the time that passed after a positive RT-PCR test are included as covariates and regressed out.

6.     The histogram of data distribution of homocysteine levels, BDI scores, and MoCA scores should be provided in the supplementary materials in the group who had recovered from COVID-19 and the control group.

7.     In the study, p<0.05 values were considered to indicate statistical significance. In my opinion, it is necessary to consider using multiple comparison correction methods, such as Bonferroni correction, Benjamini-Hochberg correction, or other types of FDR correction to control the false discovery rate and avoid erroneous conclusions.

8.     In Figure 1, the size of the text is too small to be recognized. 

Author Response

Dear Reviewer,

We are very grateful for the kind and constructive comments of yours.

We have scanned throughout the paper; thus, the necessary correction has been completed.

Thank you in advance.

Best Regards,

Reviewer 1

  1. In this study, the sample size is small and may lead to imprecise results. If possible, more patients should be included in the analysis.

This study is a prospective study. It takes a long time to increase the number of patients. In addition, the limited number of patients and the need for more comprehensive studies were stated in the limitations of our study.

  1. It is known that disruption of enzymes involved in the metabolism of B vitamins increases homocysteine concentration.Detailed information on the history of taking folic acid and the B vitamins levels in the blood needs to be provided in those who had recovered from COVID-19 and the control group, especially the history of taking folic acid.

It is known that it is possible to change homocysteine levels with the use of folic acid. For this reason, we did not include who people using folic acid in the study. It is stated as continuous drug use in the exclusion criteria. In addition, a history of folic acid use was added to the exclusion criteria.

  1. Even though there was no statistically significant difference between the two groups in terms of age, gender, and education duration, I strongly suggest that the authors should take age, gender, education duration, andthe time that passed after a positive RT-PCR test as covariates to regress out in evaluating the difference of homocysteine levels between the groups.

Regression analysis was performed as requested in both in patients who recovered from COVID-19 group, control group, and all participants. The data is written into the text and tables are added.

  1. Other demographic information in those who had recovered from COVID-19 and the control group, such as ethnicity, smoking history, alcohol consumption history, and socioeconomic status.

In the exclusion criteria, we stated already that the participants did not use alcohol or cigarettes. During the study, no information about ethnicity was obtained from the participants. It is currently not possible to take it retrospectively. Information on socioeconomic status has been added.

  1. Pearson's correlation analysis and linear regression analysis should also be performed in the control group. And I'm curious whether the results of the linear model regression for MoCA scores and homocysteine level would still be significant if age, gender, education duration, and the time that passed after a positive RT-PCR test are included as covariates and regressed out.

Pearson’s correlation and linear regression analysis were performed in the control group and added to the text and table. Linear regression analysis was performed as requested in both in patients who recovered from COVID-19 group, the control group, and all participants.The data is written into the text and tables are added.

  1. The histogram of data distribution of homocysteine levels, BDI scores, and MoCA scores should be provided in the supplementary materials in the group who had recovered from COVID-19 and the control group.

The desired histogram was made and added as supplementary material.

  1. In the study, p<0.05 values were considered to indicate statistical significance. In my opinion, it is necessary to consider using multiple comparison correction methods, such as Bonferroni correction, Benjamini-Hochberg correction, or other types of FDR correction to control the false discovery rate and avoid erroneous conclusions.

The corrections you request are made for multiple comparisons. Since there are two groups being compared here (COVID-19 and control),  these adjustments cannot be made.

  1. In Figure 1, the size of the text is too small to be recognized. 

It is corrected.

Reviewer 2 Report

This is an interesting study. The authors reported that homocysteine levels were higher in those who had recovered from COVID-19 than the control group. Those who recovered from COVID-19 had lower MoCA scores. In this subsample they found a significant negative correlation between homocysteine levels and  MoCA scores. However, it needs as substantial revision.

Major points:

1)      How did the authors select participants in the control group? If I understood correctly,  the controls were patients in the same hospital where the COVID-19 group was selected. What kind of diseases affected the control group? What are the reasons they were admitted at the neurological unit of the hospital?

2)      Please follow a scientific way to report methods and results.  Examples as follows -         The Turkish validity and reliability study of the scale was performed by Hisli in 1989. Is better to say:  Hisli [14] reported the Turkish ……. You don’t need to mention the year. Another examples: “ This scale was developed by Nasreddine et al. in 2005, and Turkish validity and reliability study was performed by Selekler et al. in 2010." 

3)      Obvious statements should be deleted. Example: “Number, percentage, mean and standard deviation were used as descriptive statistical methods in the evaluation of the data”. It is completely unnecessary. 

4)      The description of results is awkward. Examples: - "p:0.501>0.05". What does p: mean? Why say that 0.5 > 0.05? Obviously it is a NS (non-significant) result; - “ (p=0.048<0.05)”; - “There was a 0.252 positive correlation between” What do the authors mean? Is 0.252 the Pearson correlation coefficient? ; R2 is different from R2 ; - “When the correlation analyses were examined”. Please say the analyses of the correlation coefficients indicated ….; “with a value of 0.75”. Simply, r= 0.75 (F=.. df=…, P<0.001); - “the duration of education ..”. Please say years of formal education. 

5)      The authors should report the degrees of freedom. The samples used for all the tests should be described ( only COVID 19 or  COVID-19 patients + controls ?) 

6)      Do the authors check the normality of the distributions? Homoscedasticity? Assumptions to perform Pearson correlation?  Is the MoCA score a real continuous variable? 

 7)      The MoCA instrument has several limitations. This should be addressed in the discussion section. 

8)      A revision in English is necessary (scientific English).

 9)      How was the relationship between MoCA scores and homocysteine levels in the control group?

Author Response

Dear Reviewer,

We are very grateful for the kind and constructive comments of yours.

We have scanned throughout the paper; thus, the necessary correction has been completed.

Thank you in advance.

Best Regards,

Reviewer 2

This is an interesting study. The authors reported that homocysteine levels were higher in those who had recovered from COVID-19 than the control group. Those who recovered from COVID-19 had lower MoCA scores. In this subsample they found a significant negative correlation between homocysteine levels and  MoCA scores. However, it needs as substantial revision.

Major points:

1)      How did the authors select participants in the control group? If I understood correctly,  the controls were patients in the same hospital where the COVID-19 group was selected. What kind of diseases affected the control group? What are the reasons they were admitted at the neurological unit of the hospital?

Patients hospitalized in the neurology clinic due to peripheral neuronal diseases were included in the study. It was accepted that cognitive functions were not impaired in these patients. In addition, it has already been stated in the material and method section of the text that neurological diseases that can cause cognitive impairment were excluded from the study.

2)      Please follow a scientific way to report methods and results.  Examples as follows -         The Turkish validity and reliability study of the scale was performed by Hisli in 1989. Is better to say:  Hisli [14] reported the Turkish ……. You don’t need to mention the year. Another examples: “ This scale was developed by Nasreddine et al. in 2005, and Turkish validity and reliability study was performed by Selekler et al. in 2010." 

It is corrected. Painted yellow in text.

3)      Obvious statements should be deleted. Example: “Number, percentage, mean and standard deviation were used as descriptive statistical methods in the evaluation of the data”. It is completely unnecessary. 

It is corrected.

4)      The description of results is awkward. Examples: - "p:0.501>0.05". What does p: mean? Why say that 0.5 > 0.05? Obviously it is a NS (non-significant) result; - “ (p=0.048<0.05)”; - “There was a 0.252 positive correlation between” What do the authors mean? Is 0.252 the Pearson correlation coefficient? ; R2 is different from R; - “When the correlation analyses were examined”. Please say the analyses of the correlation coefficients indicated ….; “with a value of 0.75”. Simply, r= 0.75 (F=.. df=…, P<0.001); - “the duration of education ..”. Please say years of formal education. 

It is corrected. Painted yellow in text.

5)      The authors should report the degrees of freedom. The samples used for all the tests should be described ( only COVID 19 or  COVID-19 patients + controls ?) 

It is corrected

6)      Do the authors check the normality of the distributions? Homoscedasticity? Assumptions to perform Pearson correlation?  Is the MoCA score a real continuous variable? 

Normality distribution is already written in the text in the statistical analysis section. Yes, the MoCA score is a real continuous variable. For this reason, we used pearson's correlation between data.

7)      The MoCA instrument has several limitations. This should be addressed in the discussion section. 

The limitations of the MoCA score have been added to the discussion section.

8)      A revision in English is necessary (scientific English).

It is corrected. Painted yellow in text.

 9)      How was the relationship between MoCA scores and homocysteine levels in the control group?

The relationship between homocysteine and MoCA score in the control group was demonstrated by pearson’s correlation analysis.

Round 2

Reviewer 1 Report

Accept as is.